# Evaluating the Safety and Quality of Life of Colorectal Cancer Patients Treated by Autologous Immune Enhancement Therapy (AIET) in Vinmec International Hospitals

**DOI:** 10.3390/ijms231911362

**Published:** 2022-09-26

**Authors:** Hoang-Phuong Nguyen, Duc-Anh Dao Pham, Duy Dinh Nguyen, Phong Van Nguyen, Viet-Anh Bui, My-Nhung Thi Hoang, Liem Thanh Nguyen

**Affiliations:** 1Vinmec Research Institute of Stem Cell and Gene Technology, Vinmec Healthcare System, Hanoi 100000, Vietnam; 2Faculty of Biology, VNU University of Science, Vietnam National University-Hanoi, 334 Nguyen Trai, Hanoi 100000, Vietnam; 3Vinmec Times City International Hospital, Vinmec Healthcare System, 458 Minh Khai Street, Hanoi 100000, Vietnam; 4Center of Applied Science, Regenerative Medicine, and Advanced Technologies (CARA), Vinmec Healthcare System, 458 Minh Khai, Hanoi 100000, Vietnam; 5College of Health Sciences, Vin University, Hanoi 100000, Vietnam

**Keywords:** autologous immune enhancement therapy, colorectal cancer, natural killer cells, cytotoxic T lymphocytes

## Abstract

(1) Colorectal cancer (CRC) is an increasingly prevalent disease with a high mortality rate in recent years. Immune cell-based therapies have received massive attention among scientists, as they have been proven effective as low-toxicity treatments. This study evaluated the safety and effectiveness of autologous immune enhancement therapy (AIET) for CRC. (2) An open-label, single-group study, including twelve patients diagnosed with stages III and IV CRC, was conducted from January 2016 to December 2021. Twelve CRC patients received one to seven infusions of natural killer (NK)-cell and cytotoxic T-lymphocyte (CTL). Multivariate modelling was used to identify factors associated with health-related quality-of-life (HRQoL) scores. (3) After 20–21 days of culture, the NK cells increased 3535-fold, accounting for 85% of the cultured cell population. Likewise, CTLs accounted for 62.4% of the cultured cell population, which was a 1220-fold increase. Furthermore, the QoL improved with increased EORTC QLQ-C30 scores, decreased symptom severity, and reduced impairment in daily living caused by these symptoms (MDASI-GI report). Finally, a 14.3 ± 14.1-month increase in mean survival time was observed at study completion. (4) AIET demonstrated safety and improved survival time and HRQoL for CRC patients in Vietnam.

## 1. Introduction

Colorectal cancer (CRC) is the third most common cancer and the second leading cause of cancer-related deaths, with an estimated number of 1.8 million new cases and nearly 881,000 deaths according to 2018 data [1]. In Vietnam, the incidence and mortality of CRC has increased rapidly, with the total economic cost of CRC being approximately $132.9 million, representing 0.055% of the 2018 gross domestic product [2]. Most colorectal cancer patients are diagnosed with the disease when they have entered the late stages (III, IV). Cases with stage III disease are curable; however, the disease is impossible to cure when invasive cancer metastasizes to distant regions (stage IV) [3,4]. Therefore, efficient treatment methods are still being sought and developed for these cancer patients. In recent years, autologous immune enhancement therapy (AIET) has attracted the attention of many scientists. This therapy kills cancer cells directly, improving the patient’s immune system to fight against cancer cells and other comorbidities [5,6,7]. This method’s principle is to isolate immune cells from the patient’s peripheral blood and to culture one specific type of cell in their optimal medium; these cells will be stimulated to proliferate. The final step is the infusion of these expanded cells back into the patient’s body when their number and capacity meet the requirements [8,9,10].

Human natural killer (NK) cells were first discovered in 1975 when two studies by Pross and Kiessling discovered a strange group of lymphocytes, larger than T and B lymphocytes, containing different substances that are toxic to cancer cells [11,12]. Similar to other lymphocytes, NK cells originate from lymphoid progenitor cells in the bone marrow and then differentiate and are distributed into many tissues and organs in the body, such as the bone marrow, lymph nodes, spleen, peripheral blood, lung, and liver [13,14]. NKs are lymphocytes negative for CD3 and positive for the expression of CD56 at various levels [15]. Approximately 90% of NK cells are located in the peripheral blood and spleen and are cytotoxic innate lymphocytes that can lyse cancerous or virally infected cells. These cells have a lower expression of CD56 and a high expression of CD16 (CD56dimCD16+). In contrast, NK cells residing in lymph nodes exhibit high expression of CD56 but do not express CD16; these cells mediate immunoregulatory effects through the secretion of particular cytokines, such as interferon γ, in response to stimulation by IL-12, IL-15, and IL-18 [16]. Without expressing polymorphic clonotypic receptors and utilizing inhibitory receptors, NK cells can recognize abnormal expression levels of the major histocompatibility complex (MHC) class I on the surface of other cells, which helps NK cells distinguish “self” from “nonself” [17]. The balance between inhibitory signals and activating signals from several different surface receptors allows NK cells to choose their target precisely [18,19]. The indirect effects of NK cells are demonstrated through their interactions with other immune cells, such as dendritic cells (DCs), macrophages, and T cells. By producing IFN-γ, NK cells stimulate CD8⁺ T and CD4⁺ T cells to become cytotoxic T cells (CTLs). Simultaneously, cytokines secreted by NK cells stimulate B lymphocytes to produce anticancer monoclonal antibodies [20].

As CD8^+^ or killer T cells, CTLs are characterized by the expression of CD3 and CD8 (CD3^+^CD8^+^). As a crucial component of adaptive immunity, CTLs display an innate ability to destroy infected and cancerous cells through a series of mechanisms. First, by displaying the T-cell receptor (TCR) on the surface, CTLs can swiftly initiate the apoptosis processes in the target cell after specifically linking with corresponding alien antigens. Furthermore, antiviral and antitumor cytokines, such as TNF-α and IFN-γ, are secreted by CTLs. Finally, another destructive pathway by which CTLs deal with infected cells is through the Fas/FasL interaction [21,22].

By optimizing the antiviral and tumor surveillance ability of both NK cells and CTLs through serial steps from isolation and expansion, followed by reinfusion of these activated cells back to the patient body, AIET has become a promising, cutting-edge method for detecting and eliminating cancer cells [9,23,24]. This study aimed to evaluate the safety and effectiveness of AIET in twelve patients with CRC.

## 2. Results

### 2.1. Patient Characteristics

Eight male and four female patients diagnosed with CRC were enrolled in this study. All included patients had stage III and IV CRC. The median age of the patients was 59 years (ranging from 52.1 to 71.2 years old). Within this study patient cohort, seven patients exhibited tissue metastases, and five were nonmetastatic patients. The average duration of AIET after surgery was 15.7 ± 7.45 months. The physician predicted the patient’s survival time when participating in the study. This study’s average estimated survival time was 18.3 ± 13.9 months (Table 1).

### 2.2. Immune Cell Expansion Ability

Patients in this study received different numbers of immune cells and infusions, typically from 1 to 4 infusions, including five patients with one, three patients with two, one patient with three, two patients with four, and one patient with seven infusions. PBMNC isolation, immune cell extraction, and culture were executed in an ISO 14644-certified clean room. PBMNCs were isolated by Ficoll-Paque density centrifugation and then divided into two equal parts, one for NK-cell expansion and the other for CTL expansion. The patients’ immune cells were expanded by BINKIT, developed by Dr. Terunuma Hiroshi (Biotherapy Institute of Japan, Tokyo, Japan). At the seeding point of cell culture, the average percentage of NK cells was 11.6%, with a mean cell number of 2.6 × 10^6^ ranging from 0.59 to 4.46 × 10^6^. The CTL average percentage was 25%, with a mean cell number of 6.48 × 10^6,^ ranging from 1.73 to 22 × 10^6^. From day 5 of culture onwards, we observed the cells growing as clusters in the medium in both CTL and NK-cell cultures. This morphology indicated the logarithmic phase of cell proliferation (Figure 1A). The percentage (%) of CTL and NK cells also increased during the time of culture (Figure 1B).

The number of NK cells and CTLs increased significantly after 20–21 days of expansion. In the NK-cell culture, the number of cells ranged from 1.2 to 21.3 × 10^9^, with an average number of 8.8 × 10^9^, accounting for 85% of the cell population. There was an average 3535-fold increase compared to the number of NK cells at seeding. Notably, in one sample, the increase was impressively high at 9446 times (Table 2). Meanwhile, in CTL culture, the CTL numbers ranged from 0.1 to 13.3 × 10^9^, accounting for 62.4%, and increased 70- to 4847-fold with an impressively high average cell number of 6.6 × 10^9^ (Table 3). All cultures were negative for mycoplasma, bacteria, and fungi. The endotoxin level was <0.5 EU/mL.

In the patients’ peripheral blood, a significant difference was observed in the percentage of NK cells by age, with 9% for those below and 14% for those above 60 years old (*p* value = 0.027); by disease stage, with 9% for stage III patients and 14% for stage IV patients (*p* value = 0.006); and by cancer metastasis, with 14% for patients who had metastasis and 10% for nonmetastatic patients (*p* value = 0.013). A similar significant difference was observed in the percentage of CTLs by age, with 29% for patients below 60 years old and 21% for those above 60 years old (*p* value = 0.008), and by disease stage, with 28% for stage III patients and 22% for stage IV patients (*p* value = 0.020). Furthermore, the numbers of CTLs were significantly higher in stage III patients than in stage IV patients (8.58 × 10^6^ vs. 4.52 × 10^6^, *p* value = 0.005) and in metastasis-free patients than in those with metastasis (8.0 × 10^6^ vs. 4.61 × 10^6^, *p* value = 0.014). In contrast, there was no significant difference in the number of NK cells for the patients by disease stage or cancer metastasis.

After 20–21 days of culture, the total number of CTLs differed significantly by disease stage, with a higher number of CTLs in stage III patients (8259 ± 3370 × 10^6^) when compared to stage IV patients (5113 ± 3666 × 10^6^ cells) (*p* value = 0.023) and by cancer metastasis, with 4858 ± 3893 × 10^6^ cells for the patients who were detected to have metastasis and 8073 ± 3178 × 10^6^ cells for the metastasis-free patients (*p* value = 0.025). Additionally, the younger patients had a proportion of CTLs accounting for 69%, while the figure for the older patients was 56% (*p* value = 0.037). Meanwhile, the percentage of CTLs in females was 20% higher than that in males (79% vs. 59%, *p* value < 0.001). No significant difference in NK-cell expansion was detected by age, disease stage, or cancer metastasis (Table 4 and Table 5).

### 2.3. Safety of Ex Vivo Expanded Cell Infusion

The infusion was performed on twelve colorectal cancer patients with an average of 2.5 immune cell infusion infusions. During the collection of patients’ peripheral blood, no adverse events were observed, and no severe adverse events (AEs) occurred during the infusion of NK cells and CTLs.

### 2.4. Survival Time of Patients in This Study

Survival time was defined as from the date of diagnosis until death or the end of this study; the estimated survival time was provided by the doctors at the time of immune therapy. The average survival time for all patients participating in this study was 32.6 ± 16.3 months (2.72 years). Therefore, an increase in the mean survival time by 14.3 ± 14.1 months was observed at the end of the study. The results showed a statistically significant difference between the actual and estimated survival times (Figure 2). By December 2021, five patients had died; seven were alive without relapse and had a good quality of life (Table 6).

### 2.5. Quality of Life of Patients Enrolled in This Study

#### 2.5.1. Changes in Symptoms before and after Immune Cell Infusion

Table 7 compares the mean (±standard deviation) scores on MDASI-GI items before and after the infusions. Fatigue, pain, disturbed sleep, sadness, and poor appetite were the five most severe symptoms. After infusion, the patients reported that all signs were much improved. In addition, vomiting, nausea, dry mouth, and distress decreased with statistical significance (*p* < 0.05).

Regarding symptoms of colorectal cancer, except for feeling bloated, the other symptoms were significantly improved after infusions. Regarding the “reactive” dimension of symptoms, eight patients noticed significantly less impairment in general functioning, mood, work, relationships with others, and ability to walk and enjoy life.

#### 2.5.2. Changes in Patients’ Quality of Life before and after Immune Cell Infusion

Changes in the patients’ quality of life (QoL) were evaluated through the criteria of the European Organization for Research and Treatment of Cancer Quality of Life Questionnaire Version 3.0 (EORTC QLQ-C30) and MD Anderson Symptoms Inventory Gastrointestinal Cancer Module (MDASI). The EORTC is the most frequent HRQoL instrument for colorectal cancer [25]. In our study, the QoL assessment was executed before and after immune cell infusion. Unfortunately, the response rate was 67% because we only received the results from eight out of twelve patients (i.e., without patients no. 9, 10, 11, and 12). By December 2021, seven (58%) of twelve patients were relapse-free and alive with a good QoL.

All the scales and single-item QLQ-C30 measures range in scores from 0 to 100. A high scale score represents a higher response level. Notably, a high score for a functional scale or global health status represents a healthy level of functioning and a high QoL, but a high score on a symptom scale represents a high level of problems [26]. Based on the EORTC QLQ-C30, the global health status score increased from 57.3 to 71.9, and this difference was statistically significant (*p* < 0.05). The same improved result was observed across all the function scales, except for cognitive functioning.

Regarding symptom scales, significant improvement was witnessed for the dimension of symptoms for fatigue, nausea and vomiting, dyspnea, constipation, and financial difficulties (*p* < 0.05). However, symptoms such as pain, insomnia, appetite loss, and diarrhea remained a burden on the patients’ health (Table 8).

A multivariate analysis was performed to reveal the effects of patient factors on global health status and all the function scales of the EORTC QLQ-C30. We investigated the association between HRQoL and age, sex, disease stage, and survival duration (patients who survived over three years and those who survived below three years) (Table 9). Physical function before AIET therapy significantly affected survival duration. Specifically, the physical functioning score was 29.84% higher in patients who survived for over three years than in those who survived for less than three years (*p* < 0.05).

## 3. Discussion

In the last two decades, autologous immune cell therapy has become more common in supportive treatment for cancer. Autologous immune cell infusion has proven its safety for cancer treatment with an increase of 30% in efficacy when used in combination with traditional methods such as radiotherapy, chemotherapy, or surgery [27]. The vital aspect for the successful application of cellular immune therapy is an adequate number of immune cells to eradicate tumor cells without affecting other cells in the body [9]. In that regard, CTLs [28] and NK cells [29] are receiving great attention worldwide. Subramani et al. published a case report of a stage IV colon cancer patient who had already received first-line chemotherapeutic drugs with six infusions of AIET [30]. This CRC patent received an average expansion of NK cells and CTLs of 44- and 168-fold, respectively. There were no adverse reactions during this therapy and the eight-month follow-up from the first infusion of AIET. In 2014, Subramani et al. [31] reported that four cases with stage IV colon cancer received 48 × 10^6^ initial peripheral blood mononuclear cells and 2700 × 10^6^ total expanded NK and T-cell intravenous infusions. Improved prognosis, reflected by a considerable decrease in cancer markers, improved QoL, and a considerable increase in survival rates (23 months) was observed.

Using the same published protocol [32], immune cell expansion in this cohort study was performed for 12 CRC patients. Multiple intravenous infusions were administered to the patients, with a maximum of 7 infusions. We successfully achieved a maximum value for immune cell expansion at 9446-fold and 4847-fold of NK cells and CTLs, respectively, after 20–21 days of culture. These figures are equivalent to or higher than those described in other publications [30,31,33].

Our study revealed no significant difference in the NK-cell numbers at pre-expansion for the patients by age, disease stage, or cancer metastasis. The CTL numbers in stage III and nonmetastatic patients were significantly higher than those in stage IV and metastatic patients in both pre and after-culture. Therefore, the expansive capacity of T immune cells depends on the cancer stage and metastatic status.

Previous studies have reported various doses of autologous NK cells and CTLs for cancer treatment [32,34,35]. Our study provides evidence of the excellent tolerability of AIET for colorectal cancer patients. None of the patients in this study suffered any serious AEs from AIET. Similarly, the safety of the intravenous infusion of immune cells into cancer patients was previously reported [36,37,38].

Investigation of CRC patients’ QoL is essential for evaluating chronic or late effects of the disease and treatment and adjusting treatment strategies to patients’ needs [25]. Our study also demonstrated that the patients had a steady improvement in QoL, symptom severity, and symptom impact. Global health status and all aspects of functional dimensions (physical, role, emotional, social, and cognitive) improved over time according to The European Organization for Research and Treatment of Cancer Quality of Life Questionnaire (EORTC QLQ-C30). Furthermore, some significant improvements in pain, fatigue, lack of appetite, sadness, mood, and enjoyment of life were observed according to the MDASI-GI. The high QoL scores and symptom improvement after the follow-up period could be explained by the fact that autologous immune cell transplantation positively effected changes in the patients’ mental, physical, and social characteristics. Our results are consistent with other reports, indicating a significant improvement in the patients’ QoL during follow-up [33,39,40,41].

QoL is not only important for the well-being of cancer patients but also influences survival and response to therapy. Some authors have proposed that QoL is also an independent predictor of survival and response to therapy in cancer patients [42,43]. In our study, the multivariate analysis showed that the physical function score before AIET was 29.84% higher in patients who survived over three years than in those who survived below three years. Our finding that physical function scores correlate with better survival in CRC is consistent with studies of localized head and neck and oesophagogastric cancer patients [44,45]. In addition, this finding was reported by Braun et al. in 396 stage III-IV CRC patients, where a lower risk of death was associated with a 10-point improvement in physical function three months after treatment (HR, 0.86; 95% CI, 0.78 to 0.94; *p* = 0.001) [46]. This finding is vital for suggesting that baseline QoL should be considered when planning treatment, and regular QoL evaluations should be performed during the follow-up.

Autologous immune cell transplantation has received tremendous attention and has been deemed a promising method to increase disease-free survival [40]. The mean survival time for the patients in our study was 32.6 months after the transfusion; hence, it increased their survival approximately 1.8 times compared to the estimated survival time at the point of enrolment. This result was relatively higher than that in other publication [31,33,41]. The 1-year survival rate of our study was 83%, which was higher than that in another published report [47]. By December 2021, five patients had died; seven were alive without relapse and had a good QoL. Thus, immune enhancement may be a solution to increase CRC survival rates.

Our study still has limitations. The number of patients was small and, more importantly, we lacked a comparative/control group. In further work, we plan to perform a follow-up study with a control group so that we will have a more accurate conclusion. Besides, the biomarkers should be determined to predict the better response of the treatment. In the past ten years, immune checkpoint inhibitors have gained a lot of attention since they have showed great ability in the improvement of the outcome of cancer patients [48]. PD-L1 expression has been demonstrated as a predictive biomarker for sensitivity to cancer treatment, especially to immune checkpoint inhibitor [49,50]. We will also measure the level of PD-L1 in the cancer patients using AIET in future works.

## 4. Materials and Methods

### 4.1. Patients

#### 4.1.1. Inclusion Criteria

-Patients were 18–75 years of age.-Patients had been diagnosed with colorectal cancer.-Patients signed the written informed consent form.

#### 4.1.2. Exclusion Criteria

Severe health conditions such as serious infection, autoimmune diseases, or the use of any antirejection drugs.

### 4.2. Study Design

An open-label uncontrolled phase I clinical trial was performed.

### 4.3. Research Setting and Duration

The study was carried out at the Oncology Department, Vinmec Times City International Hospital, and Vinmec Central Park International Hospital from January 2016 to December 2021. Patients provided written informed consent, and the study was approved by the Ethics Committee of the Vinmec International Hospital (document no. 28/2022/CN-HDDD VMEC).

**Trial registration:**ClinicalTrials.gov identifier: NCT05520372. Name of the registry: Vinmec Research Institute of Stem Cell and Gene Technology. https://clinicaltrials.gov/ct2/show/NCT05520372. Registered on 29 August 2022. The trial results will also be published according to the CONSORT statement at conferences and reported in peer-reviewed journals.

### 4.4. Cohort Size

During the study period, twelve patients met the inclusion and exclusion criteria. The peripheral blood of twelve patients was collected before the start of radiotherapy. Twelve peripheral blood samples were marked as Patient 1 (PT 1) to PT 12, corresponding to the twelve patients. The expanded immune cells suspended in 70 mL of solution were transfused intravenously approximately 1 h before and after radiotherapy.

### 4.5. Isolation and Expansion of NK Cells and CTLs from Peripheral Blood

In brief, peripheral blood mononuclear cells (PBMNCs) were collected by density gradient centrifugation using Ficoll-Paque (GE Healthcare, Uppsala, Sweden) and were then divided into two equal parts: one for NK-cell expansion and the other for CTL expansion. Both were cultured using BINKIT^®^ (Biotherapy Institute of Japan, Tokyo, Japan) at a density of 1 × 10^6^ cells/mL in the initial cell medium supplemented with 5% heat-inactivated autologous plasma. The one for NK-cell expansion was cultured in an anti-cluster of differentiation 16 (CD16) monoclonal antibody-immobilized culture flask, while the other was cultured in an anti-CD3 monoclonal antibody-immobilized flask for CTL expansion. After three days, the culture medium was changed and subcultured every 2–3 days in subculture medium, then supplemented with 5% heat-inactivated autologous plasma to maintain a concentration of 0.8–1.0 × 10^6^ cells/mL without discarding the old medium. When the number of cells increased logarithmically, the cultured cells were transferred into culture bags (Nipro, Osaka, Japan) until the end of the culture. The cell processing center was set up in compliance with Good Manufacturing Practice (GMP) standards. The cells had to be administered within 14 h of cell processing, so they were transported from the facility promptly for infusion to assure optimal viability.

The phenotypes of expanded cells and PBMNCs at baseline (day 0) and the end of culture were analyzed by flow cytometry. Monoclonal antibodies specific for CD3, CD8, CD56, and CD4 were conjugated with Pacific Blue, fluorescein isothiocyanate (FITC), R Phycoerythrin (PE), and Allophycocyanin-Alexa Fluor 750 (APC-Alexa Fluor 750), respectively, and the corresponding isotypes were used for the characterization of cell populations. Cells were analyzed by a Navios Cytometer (Beckman Coulter, Brea, CA, USA), and data were acquired by Navios software, version 3.2, according to the manufacturer’s instructions.

### 4.6. Dosage and Duration

The total expanded immune cells in one AIET infusion was transfused into the patients on a case-by-case basis. Doctors determined the number of infusions depending on the severity/stage/metastasis and the patient’s general health. One AIET infusion consisted of one NK-cell and one CTL infusion. Each AIET infusion required 50 mL of peripheral blood. For optimal efficacy, immune cell treatments were given alone or in combination with other conventional treatments, such as chemotherapy or radiotherapy. This therapy needed to be discontinued three days before the immune cell infusion and resumed three days after the infusion for patients undergoing chemotherapy. Peripheral blood had to be collected before the start of radiotherapy. Approximately 70 mL of expanded immune cell solution was transfused intravenously on the infusion day within 15–60 min at one hour before or after radiotherapy. In addition, the patients were advised to remain hospitalized for 8 h for observation.

### 4.7. Quality of Life Assessment

Evaluating QoL has become more common in oncology, particularly cancer therapy trials [51,52]. QoL was evaluated by the European Organization for Research and Treatment of Cancer Quality of Life Questionnaire Version 3.0 (EORTC QLQ-C30) [53], which includes 30 items describing global health status, functional dimensions, symptom dimensions, and financial difficulties. The functional dimensions are physical, emotional, cognitive, and social. The symptom dimensions consist of fatigue, nausea, vomiting, pain, dyspnea, insomnia, appetite loss, constipation, and diarrhea. Scores from each item were converted to a scale from 0–100. The higher the score was for a functional scale and global health status, the better the body functions and overall health status were. However, the higher the symptom scale score was, the worse the patient’s health was affected by that symptom. A scoring procedure was applied according to the EORTC QLQ-C30 Scoring Manual [26].

In addition, symptoms and the subsequent interference with the patients’ daily living activities were rated using the MD Anderson Symptom Inventory Gastrointestinal Cancer Module (MDASI-GI) [54]. The MDASI-GI is psychometrically validated and contains a 24-item questionnaire. MDASI-GI symptom items are assessed on a numeric scale ranging from 0 or “not present” to 10 or “as bad as you can imagine”.

### 4.8. Statistical Analysis

Data were analyzed using R Studio software version 1.4.1106 (RStudio, Boston, MA, USA). Descriptive statistics included the frequency, percentage, mean, and standard deviation to describe the research subjects’ characteristics of cells. The nonparametric paired Wilcoxon signed rank test was used to compare values before and after the intervention of the same group. The total score and each dimension’s score of the European Organization for Research and Treatment of Cancer Quality of Life Questionnaire Version 3.0 (EORTC QLQ-C30) and MD Anderson Symptoms Inventory Gastrointestinal Cancer Module (MDASI) were described by the mean (SD). The preintervention and postintervention results were compared using the bootstrap method, which may be more appropriate for analyzing HRQoL data than a conventional statistical method [55,56,57]. The survival curves were generated using the Kaplan-Meier model. The difference between two or more mean values was considered statistically significant when *p* < 0.05.

## 5. Conclusions

To conclude, we successfully expanded immune cells from the peripheral blood of twelve patients with CRC. The numbers and quality of expanded immune cells met the clinical requirements. The present study demonstrated that this therapy is safe and may improve the QoL for such patients.

## Figures and Tables

**Figure 1 ijms-23-11362-f001:**
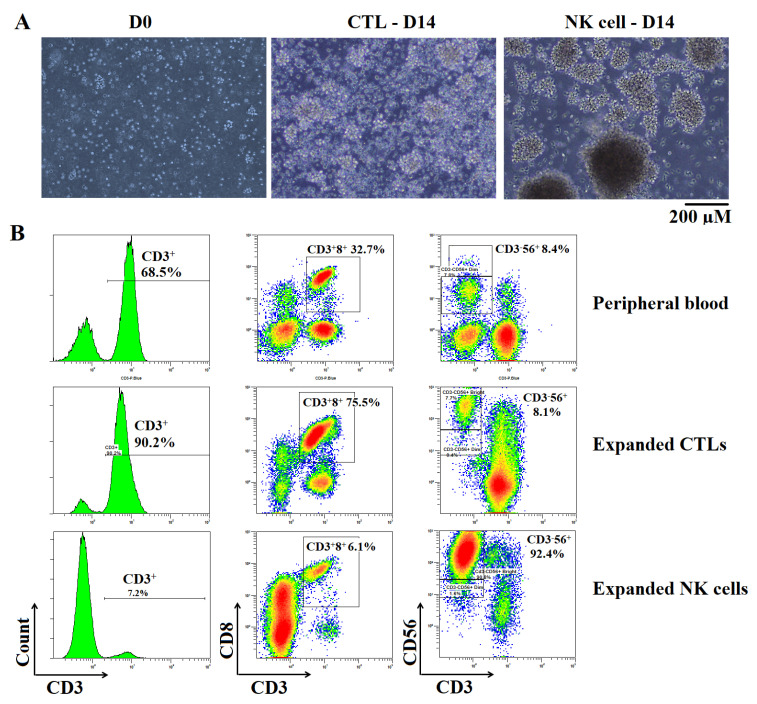
The expansion of CTL and NK cells. (**A**) The morphology of CTL and NK cells at D0 and D14 of culture. Note that the cells grew as clusters when they were in the logarithmic phase. (**B**) The immune phenotypes of CTLs and NK cells at peripheral blood and after expansion.

**Figure 2 ijms-23-11362-f002:**
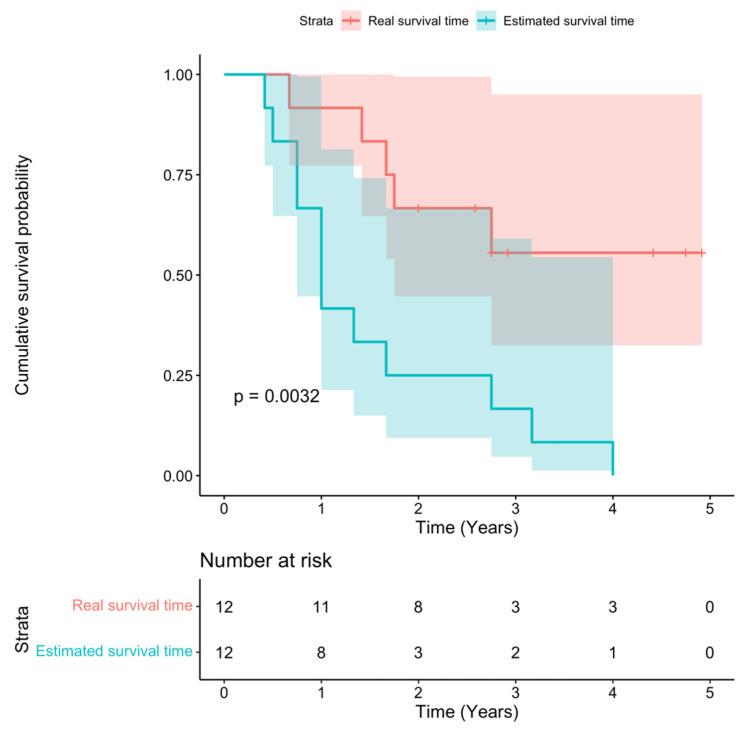
Survival time of patients enrolled in this study.

**Table 1 ijms-23-11362-t001:** Details of the patients enrolled in this study.

Patient	Sex	Age	Stage	Metastatic Site	Pretreatment	Duration of AIET after Surgery (Months)	Estimated Survival Prior to AIET (Months)
PT 1	M	54.5	IV	M1 m	-	14	12
PT 2	F	58.7	IV	M1 m (liver)	Chemo	8	6
PT 3	M	54.6	III	-	Chemo	19	48
PT 4	F	59.7	III	-	Surgical/Chemo	17	20
PT 5	M	66.8	III	-	Surgical/Chemo	20	38
PT 6	F	52.1	III	-	Surgical/Chemo	2	9
PT 7	M	62.5	IV	M1 m	Chemo	17	9
PT 8	M	65.7	IV	M1 m (liver)	Chemo	4	12
PT 9	M	58.3	III	-	Chemo	23	16
PT 10	M	62.1	IV	M1 m (liver, lung)	-	16	5
PT 11	F	71.2	IV	M1 m	Surgical/Chemo	24	33
PT 12	M	54.3	IV	M1 m	Surgical/Chemo	24	12
Mean (SD) Median [Min; Max] *	-	60.0 (5.83) 59.2 [52.1; 71.2]	-	-	-	15.7 (7.45) 17.0 [2; 24]	18.3 (13.9) 12.0 [5.0; 48.0]

* Mean, min, and max values were calculated from twelve patients.

**Table 2 ijms-23-11362-t002:** Average numbers of NK cells pre- and post-culture.

Patient	Number of Infusions	Number of NK Cells/Infusions (×10^6^) before Expansion (Mean ± SD)	Number of NK Cells/Infusions (×10^6^) after Expansion (Mean ± SD)	Fold Increase in NK Cells (Mean ± SD)
PT 1	2	2.04 ± 0.22	12,189 ± 2438	5931 ± 534
PT 2	1	3.71	3555	959
PT 3	3	1.14 ± 0.61	4891.6 ± 1829.4	4646 ± 1311.7
PT 4	1	0.59	3215.2	5448.5
PT 5	7	3.34 ± 0.86	11,575.4 ± 6978.7	1745.5 ± 3892
PT 6	2	2.4 ± 0	2903 ± 2372.4	1211 ± 989.4
PT 7	2	2.72 ± 1.52	14,903.4 ± 883.1	6604 ± 4019.7
PT 8	4	2.70 ± 0.68	6742.7 ± 990.1	2687 ± 1029.4
PT 9	1	3.86	13,043.5	3382
PT 10	1	1.90	1611	849
PT 11	1	3.38	5014.6	1485
PT 12	4	2.96 ± 0.55	11,525.8 ± 5631.4	3862 ± 1452.6

**Table 3 ijms-23-11362-t003:** Average numbers of CTLs pre- and post-culture.

Patient	Number of Infusions	Number of CTLs/Infusions (×10^6^) before Expansion (Mean ± SD)	Number of CTLs/Infusions (×10^6^) after Expansion (Mean ± SD)	Fold Increase in CTLs (Mean ± SD)
PT 1	2	6.66 ± 2.75	10,148.2 ± 1747.5	1606 ± 402.8
PT 2	1	8.23	6016.3	731
PT 3	3	12.98 ± 7.98	9696.7 ± 4235.4	1036 ± 705.8
PT 4	1	5.80	11,563.9	1995.2
PT 5	7	7.88 ± 2.07	7463 ± 3574.1	967 ± 467.8
PT 6	2	6.43 ± 0	7514.4 ± 3479.4	1168 ± 540.7
PT 7	2	3.92 ± 1.85	6770.3 ± 55.97	1950 ± 939
PT 8	4	1.91 ± 0.20	4097.1 ± 4881.6	1966 ± 2327.8
PT 9	1	7.29	7699	1056.7
PT 10	1	2.34	1109	474
PT 11	1	3.99	8108	2034.1
PT 12	4	6.11 ± 0.50	2809.3 ± 275.7	477 ± 275.7

**Table 4 ijms-23-11362-t004:** The relative relationship of pre-expanded immune cells with the patient’s age, disease stage, and cancer metastasis ^#^.

		Preexpanded Immune Cells			
		Percentage of NK Cell (CD3^−^/56^+^) (%)	NK Cell Number (×10^6^)	Percentage of CTL (CD3^+^/8^+^) (%)	CTL Number (×10^6^)
**Age group**	Below 60 (*n* = 14) (Mean ± SD)	9 ± 6	2.31 ± 1.07	29 ± 7	7.92 ± 4.28
	Above 60 (*n* = 15) (Mean ± SD)	14 ± 3	2.99 ± 0.88	21 ± 8	5.13 ± 3.12
	*p* value	0.027	0.073	0.008	0.058
**Stage**	III (*n* = 14) (Mean ± SD)	9 ± 4	2.57 ± 1.27	28 ± 5	8.58 ± 4.24
	IV (*n* = 15) (Mean ± SD)	14 ± 4	2.74 ± 0.75	22 ± 9	4.52 ± 2.36
	*p* value	0.006	0.668	0.020	0.005
**Cancer metastatic**	Yes (*n* = 13) (Mean ± SD)	14 ± 5	2.75 ± 0.68	22 ± 10	4.61 ± 2.48
	No (*n* = 16) (Mean ± SD)	10 ± 4	2.59 ± 1.25	27 ± 6	8.0 ± 4.28
	*p* value	0.013	0.674	0.104	0.014

^#^ Values were calculated from a total of 29 infusions.

**Table 5 ijms-23-11362-t005:** The relative relationship of post-expanded immune cells with the patient’s age, disease stage, and cancer metastasis ^#^.

		Percentage of NK Cell (CD3^−^/56^+^) (%)	CTL Number (×10^6^)	Fold Increase of NK Cell	Percentage of CTL (CD3^+^/8^+^) (%)	CTL Number (×10^6^)	Fold Increase of CTL
**Age group**	Below 60 (*n* = 14) (Mean ± SD)	79 ± 22	7913 ± 5168	3818 ± 1894	3818 ± 1894	7209 ± 3837	1025 ± 596
	Above 60 (*n* = 15) (Mean ± SD)	91 ± 12	9629 ± 5958	3271 ± 2240	56 ± 17	6093 ± 3846	1403 ± 1279
	*p* value	0.086	0.414	0.482	0.037	0.441	0.315
**Stage**	III (*n* = 14) (Mean ± SD)	81 ± 23	8412 ± 6330	3426 ± 1782	66 ± 15	8259 ± 3370	1091 ± 522
	IV (*n* = 15) (Mean ± SD)	90 ± 12	9163 ± 4932	3637 ± 2352	59 ± 20	5113 ± 3666	1342 ± 1325
	*p* value	0.224	0.726	0.786	0.328	0.023	0.505
**Cancer metastatic**	Yes (*n* = 13) (Mean ± SD)	89 ± 13	8280 ± 4687	3181 ± 1848	58 ± 21	4858 ± 3893	1248 ± 1379
	No (*n* = 16) (Mean ± SD)	82 ± 22	9223 ± 6301	3823 ± 2238	66 ± 14	8073 ± 3178	1198 ± 618
	*p* value	0.297	0.648	0.405	0.265	0.025	0.905

^#^ Values were calculated from a total of 29 infusions.

**Table 6 ijms-23-11362-t006:** The survival time at last evaluation, patient status, and causes of death.

Patient	Survival Time at Last Evaluation (Months)	Relapse	Vital Status (Until December 2021)	Causes of Death (If Any)
PT 1	33	Yes (May 2020)	Died (October 2021)	Pneumonia, respiratory failure
PT 2	57	No	Alive	
PT 3	59	No	Alive	
PT 4	21	Yes (July 2019	Died (2020)	Circulatory failure
PT 5	53	No	Alive	
PT 6	24	No	Alive	
PT 7	35	No	Alive	
PT 8	20	Yes (February 2021)	Died (June 2021)	Respiratory failure, tumor invasion of the chest wall, tumor compression of the nerve (lung metastases).
PT 9	31	No	Alive	
PT 10	8	Yes (May 2020)	Died (June 2020)	End-stage colon cancer, dark blood in the stool, multiorgan failure.
PT 11	33	No	Alive	
PT 12	17	Yes (July 2020)	Died (December 2020)	Malignant bowel obstruction, abdominal tumors, tumor compression of vital abdominal organs.

**Table 7 ijms-23-11362-t007:** Symptom improvements according to the MDASI-GI.

Criteria	Before Infusion (*n* = 8) Mean (SD) Median [Min; Max]	After Infusion (*n* = 8) Mean (SD) Median [Min; Max]	Change in the MDASI-GI Score (Mean Difference) Mean [95% CI]
*Symptom severity*			
Pain	4.38 (2.50) 4.50 [1.00; 9.00]	2.63 (2.72) 2.00 [0; 8.00]	1.75 * [1.25; 2.25]
Fatigue	4.88 (2.95) 5.50 [1.00; 9.00]	2.13 (1.55) 1.50 [1.00; 5.0]	2.75 * [1.38; 4.13]
Nausea	2.88 (3.00) 2.50 [0; 9.00]	0.875 (1.13) 0.50 [0; 3.00]	2.0 * [0.5; 3.87]
Disturbed sleep	3.63 (2.62) 3.50 [0; 7.0]	1.38 (1.69) 0.50 [0; 4.0]	2.25 * [0.75; 4.0]
Distress	2.50 (2.33) 2.50 [0; 7.0]	1.25 (0.89) 1.00 [0; 3.00]	1.13 * [0; 2.75]
Shortness of breath	0.250 (0.463) 0 [0; 1.00]	0.250 (0.463) 0 [0; 1.00]	0 [−0.38; 0.38]
Impaired memory	0.875 (0.835) 1.00 [0; 2.00]	0.750 (0.707) 1.00 [0; 2.00]	0.13 [−0.25; 0.50]
Lack of appetite	3.63 (2.45) 3.00 [1.00; 7.00]	0.875 (0.835) 1.00 [0; 2.00]	2.75 * [1.25; 4.25]
Drowsiness	0.875 (1.73) 0 [0; 5.00]	1.13 (1.89) 0 [0; 5.00]	−0.25 [−1.13; 0.25]
Dry mouth	1.25 (1.67) 1.00 [0; 5.0]	0.13 (0.35) 0 [0; 1.00]	1.13 * [0.25; 2.38]
Sadness	4.00 (2.7) 4.00 [0; 8.0]	1.38 (1.77) 1.0 [0; 5.0]	2.63 * [1.13; 4.50]
Vomiting	2.38 (2.07) 2.50 [0, 6.00]	0.63 (1.06) 0 [0, 3.00]	1.75 * [0.63; 3.13]
Numbness/tingling	0.13 (0.35) 0.13 (0.35)	0.13 (0.35) 0 [0; 1.00]	-
*Specific symptoms for gastrointestinal cancer*			
Constipation	1.50 (1.93) 0.500 [0; 5.00]	0 [0, 1.00] 0.500 [0; 3.00]	0.75 * [0; 2.0]
Diarrhea or watery stools	0.500 (1.07) 0 [0; 3.00]	0.250 (0.463) 0 [0; 1.00]	0.25 * [0; 0.75]
Difficulty swallowing	1.38 (2.00) 0.500 [0; 5.00]	0.375 (0.744) 0 [0; 2.00]	0.88 * [0.13; 2.25]
Change in taste	3.50 (3.07) 3.00 [0; 9.00]	1.00 (0.756) 1.00 [0; 2.00]	2.5 * [0.63; 4.25]
Feeling bloated	0.875 (1.73) 0 [0; 5.00]	1.00 (1.77) 0 [0; 5.00]	−0.13 [−1.88; 1.13]
*Symptom interference*			
General activity	4.00 (2.67) 4.00 [0; 8.00]	2.00 (2.00) 2.00 (2.00)	2.0 * [0.75; 3.25]
Mood	3.75 (2.55) 4.00 [0; 8.00]	1.75 (2.19) 1.00 [0; 7.00]	2.0 * [1.0; 3.0]
Work	3.25 (2.55) 4.00 [0; 6.00]	1.63 (1.69) 1.00 [0; 5.00]	1.63 * [0.38; 3.0]
Relations with others	2.75 (1.98) 3.00 [0; 5.00]	1.63 (1.69) 1.00 [0; 5.00]	1.13 * [0.25; 2.0]
Walking	3.00 (2.56) 3.00 [0; 6.00]	1.38 (1.41) 1.00 [0; 4.00]	1.63 * [0.5; 3.0]
Enjoyment of life	4.25 (2.55) 4.50 [0; 8.00]	2.00 (1.77) 1.00 [0; 5.00]	2.25 * [1.0; 3.50]

* Bootstrap *p*-value < 0.05.

**Table 8 ijms-23-11362-t008:** Changes in quality of life according to the EORTC QLQ-C30.

Criteria	Before Infusion (*n* = 8) Mean (SD) Median [Min; Max]	After Infusion (*n* = 8) Mean (SD) Median [Min; Max]	Change in the QoL Score (Mean Difference) Mean [95% CI]
*Global health status*	57.3 (18.1) 50.0 [33.3; 83.3]	71.9 (13.3) 75.0 [41.7; 83.3]	13.5 * [4.17; 26.07]
*Function scales*			
Physical functioning	62.5 (19.5) 60.0 [40.0; 93.3]	76.7 (16.7) 86.7 [46.7; 93.3]	13.3 * [5.83; 25.00]
Role functioning	62.5 (21.4) 58.3 [33.3; 100]	77.1 (17.7) 75.0 [50.0; 100]	14.58 * [4.17; 29.17]
Emotional functioning	70.8 (21.4) 66.7 [33.3; 100]	90.6 (15.1) 100 [66.7; 100]	19.79 * [8.33; 29.17]
Cognitive functioning	89.6 (15.3) 100 [66.7; 100]	95.8 (14.8) 100 [66.7; 117]	6.25 [−2.08; 16.67]
Social functioning	68.8 (24.3) 66.7 [33.3; 100]	81.3 (22.6) 91.7 [50.0; 100]	12.5 * [0; 25.0]
*Symptom scales*			
Fatigue	36.1 (24.3) 38.9 [0; 66.7]	23.6 (21.8) 27.8 [0; 55.6]	12.5 * [1.39; 2 25.03]
Nausea and vomiting	20.8 (17.3) 33.3 [0; 33.3]	6.25 (17.7) 0 [0; 50.0]	14.6 * [2.03; 29.17]
Pain	41.7 (26.7) 41 [0; 83.3]	29.2 (34.2) 25.0 [0; 100]	11.9 [−1.25 × 10^−9^; 2.71 × 10^1^]
Dyspnea	8.33 (15.4) 0 [0; 33.3]	4.17 (11.8) 0 [0; 33.3]	4.17 * [0; 12.50]
Insomnia	33.3 (17.8) 33.3 [0; 66.7]	20.8 (24.8) 16.7 [0; 66.7]	12.50 [−8.33; 33.33]
Appetite loss	25.0 (15.4) 33.3 [0; 33.3]	16.7 (25.2) 0 [0; 66.7]	8.33 [−12.5; 25.0]
Constipation	16.7 (25.2) 0 [0; 66.7]	8.33 (15.4) 0 [0; 33.3]	8.33 * [0; 2.50]
Diarrhea	4.17 (11.8) 0 [0; 33.3]	4.17 (11.8) 0 [0; 33.3]	0
Financial difficulties	41.7 (29.5) 50.0 [0; 66.7]	33.3 (30.9) 33.3 [0; 66.7]	8.33 * [0; 20.83]

* Bootstrap *p*-value < 0.05.

**Table 9 ijms-23-11362-t009:** Multivariate analysis of global health status before AIET.

Variable ^a^		% Difference											
		General QoL		General Function									
	Unit or Baseline Group	Global	*p*-Value	Physical	*p*-Value	Role	*p*-Value	Emotional	*p*-Value	Cognitive	*p*-Value	Social	*p*-Value
**Age**	5 years	−9.52	0.15	−1.35	0.67	5.07	0.39	8.93	0.16	3.92	0.58	8.61	0.41
**Sex**	Female	−11.87	0.34	−15.74	0.08	−18.42	0.18	−10.86	0.37	−11.50	0.46	−15.55	0.47
**Stage**	IV	−13.73	0.29	−10.11	0.21	−20.82	0.15	−20.38	0.14	−0.02	0.99	−18.31	0.41
**Survival duration**	>3 years	15.58	0.24	29.84	0.01 *	21.80	0.14	22.11	0.12	17.40	0.30	19.98	0.38

^a^ All variables mutually adjusted in the model. * *p* < 0.05.

## Data Availability

The data that support the findings of this study are available from the corresponding author, Liem NT, upon reasonable request.

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
