# Peer review of "Evaluating the Safety and Quality of Life of Colorectal Cancer Patients Treated by Autologous Immune Enhancement Therapy (AIET) in Vinmec International Hospitals"

_ijms, 2022, doi:10.3390/ijms231911362_

Round 1

Reviewer 1 Report

The current manuscript is a description of valuable observation collected during the therapy of colorectal patients. The therapy introduced for these patients was based on the immune effector cells to improve the overall parameters. This procedure was found to be effective and that gives hope for patients treated in other hospitals… however, the manuscript is quite confusing. Authors should rearrange the text to highlight the main message since it is hidden below too many information.

The detailed remarks are presented below:

Introduction:

Lines 40-41: „Therefore, novel cancer treatment methods are considered the light at the end of the tunnel for these cancer patients.” – here sth is missing: “… are considered as the light…”??

Results:

The word: “sitting” is quite confusing? It is the same as every single sample collected from patients?

The figures are not consistent with figure legends, for instance: on Figure 1 the flow dotplots are described as ”peripheral blood”, “expanded CTLs” and “expanded NK”, whereas the legend stands “The immune phenotypes of CTLs and NK cells pre- and postculture.”. for random, unexperienced reader is better (more clear) when both parts are coherent.

Lines 91-92: “At the point of enrolment, the average estimated survival time in this study was 18.3 ± 13.9 months.” – this sentence is confusing. “estimated survival time” from diagnosis till enrolment into study?

Lines 103-105: “At the seeding point of cell culture, the percentage of NK cells was 11.6%, with a mean of 2.6 × 106 ranging from 0.59 to 4.46 × 106. The CTL percentage was 25%, with a mean of 6.48 × 106, ranging from 1.73 to 22 × 106.” – Where these values (11.6% and 25% for NK and CTLs, respectively) comes from? The ranges given next are different.

Line 111: “the number of cells ranged from 1,225.5 to 21,315.7 × 106” – the samples with only 1 mln of cells had the same value for the following therapy? Did You consider the exclusion of such samples?

Did You check what types of cells were is Your samples, except NK and CTLs?

Table 4/3 – is prepared very confusing. What values are put is brackets? In my opinion it is not necessary to repeat data from table in the text. Consider rearrangement the manuscript to choose the best way for data presentation.

The Discussion paragraph mostly repeats results and this is not necessary.

The manuscript in spite of few remarks is valuable and interesting. It require a review to be considered for publication.

Author Response

Response to Reviewer 1 Comments

Introduction:

Point 1: Lines 40-41: „Therefore, novel cancer treatment methods are considered the light at the end of the tunnel for these cancer patients.” – here sth is missing: “… are considered as the light…”??

We agree with the reviewer's comment. Therefore, we have added information according to the reviewer's suggestion as the following:

"Therefore, efficient treatment methods are still seeking and developing for these cancer patients."

Results:

Point 2: The word: “sitting” is quite confusing? It is the same as every single sample collected from patients? 

We changed “sitting” to “infusion” for more clarity in the abstract (line 20), results (lines 97-99), Table 2, Table 3, and method parts (lines 168, 352-355), accordingly.

Point 3: The figures are not consistent with figure legends, for instance: on Figure 1 the flow dotplots are described as ”peripheral blood”, “expanded CTLs” and “expanded NK”, whereas the legend stands “The immune phenotypes of CTLs and NK cells pre- and postculture.”. for random, unexperienced reader is better (more clear) when both parts are coherent. 

We changed the figure legend of Figure 1 into “The immune phenotypes of CTLs and NK cells at peripheral blood and after expansion” to make it coherent to the figure.

Point 4: Lines 91-92: “At the point of enrolment, the average estimated survival time in this study was 18.3 ± 13.9 months.” – this sentence is confusing. “estimated survival time” from diagnosis till enrolment into study? 

In this study, the physician predicted the patient's survival time when participating in the study. We rewrote the sentence to make it clearer (lines 91-93).

Point 5: Lines 103-105: “At the seeding point of cell culture, the percentage of NK cells was 11.6%, with a mean of 2.6 × 106 ranging from 0.59 to 4.46 × 106. The CTL percentage was 25%, with a mean of 6.48 × 106, ranging from 1.73 to 22 × 106.” – Where these values (11.6% and 25% for NK and CTLs, respectively) comes from? The ranges given next are different.

In this sentence, we mentioned both the percentage and the number of immune cells at the seeding point. However, the sentence was not clear. We rewrote the sentence to “At the seeding point of cell culture, the average percentage of NK cells was 11.6%, with a mean cell number of 2.6 × 106 ranging from 0.59 to 4.46 × 106. The CTL average percentage was 25%, with a mean cell number of 6.48 × 106, ranging from 1.73 to 22 × 106” (lines 104-107).

Point 6: Line 111: “the number of cells ranged from 1,225.5 to 21,315.7 × 106” – the samples with only 1 mln of cells had the same value for the following therapy? Did You consider the exclusion of such samples? 

In fact, the number of expanded cells was from 1.2 – 21.3 billion cells. The way we performed the number may make it confusing. The sentence was rewritten from 1.2 to 21.3 × 109 (for NK cells), and from 0.1 to 13.3 × 109 (for CTLs) (lines 112-117).

Point 7: Did You check what types of cells were is Your samples, except NK and CTLs?

In this study, we focused on NK cells and CTLs after expansion, hence we used the panel of CD3,4,8 and 56 for the immune cell phenotyping. The expanded NK cell population had higher purity with 85% NK cells, the remaining were CD3+ cells. The expanded CTL population had lower purity with 62.4% CTLs, the remaining were also CD3+ cells. Our results were consistent with a study published by Prof. Terunuma H team developing the AIET kit, that the remaining population in NK expansion was CD3+ T cells (consisting of CD3+Vγ9- and CD3+Vγ9+), and the remaining population in CTL expansion was mostly CD3+Vγ9+ cells [1]. Besides, in our published paper about immune cell expansion in lung cancer patients, we also indicated that the remaining population in NK cell expansion mostly was CD3+CD56+ NKT cells, which are also cells responsible for anticancer effect [2].

  1. Deng, X., Terunuma, H., Terunuma, A., Takane, T., & Nieda, M. (2014). Ex vivo-expanded natural killer cells kill cancer cells more effectively than ex vivo-expanded γδ T cells or αβ T cells. International Immunopharmacology, 22(2), 486–491. doi:10.1016/j.intimp.2014.07.036
  2. Nhung, H.T.M.; Anh, B.V.; Huyen, T.L.; Hiep, D.T.; Thao, C.T.; Lam, P.N.; Liem, N.T. Ex Vivo Expansion of Human Peripheral Blood Natural Killer Cells and Cytotoxic T Lymphocytes from Lung Cancer Patients. Oncol. Lett. 2018, 15, 5730–5738, doi:10.3892/OL.2018.8029.

Point 8: Table 4/3 – is prepared very confusing. What values are put is brackets? In my opinion it is not necessary to repeat data from table in the text. Consider rearrangement the manuscript to choose the best way for data presentation. 

Based on the reviewer's comment, Table 4 was rearranged and removed the relationship between immune cells and the patient’s sex as below.

Table 4a: The relative relationship of pre-expanded immune cells with the patient’s age, disease stage and cancer metastasis#

Preexpanded immune cells

Percentage of NK cell(CD3-/56+)(%)

NK cell number (x106)

Percentage of CTL (CD3+/8+) (%)

CTL number(x106)

Age group

Below 60
(N=
14)

(Mean ±SD)

9 ± 6

2.31 ± 1.07

29 ± 7

7.92 ± 4.28

Above 60
(Mean ±SD)

(N=15)

14 ± 3

2.99 ± 0.88

21 ± 8

5.13 ± 3.12

P value

0.027*

0.073

0.008*

0.058

Stage

III

(Mean ±SD)

(N=14)

9 ± 4

2.57 ± 1.27

28 ± 5

8.58 ± 4.24

IV
(Mean ±SD)

(N=15)

14 ± 4

2.74 ± 0.75

22 ± 9

4.52 ± 2.36

P value

0.006

0.668

0.020

0.005

Cancer metastatic

Yes

(Mean ±SD) (N=13)

14 ± 5

2.75 ± 0.68

22 ± 10

4.61 ± 2.48

No

(Mean ±SD) (N=16)

10 ± 4

2.59 ± 1.25

27 ± 6

8.0 ± 4.28

P value

0.013

0.674

0.104

0.014

# Values were calculated from a total of 29 infusions

Table 4b: The relative relationship of post-expanded immune cells with the patient’s age, disease stage and cancer metastasis#

Percentage of NK cell(CD3-/56+)(%)

CTL number(x106)

Fold increase of NK cell

Percentage of CTL (CD3+/8+) (%)

CTL number(x106)

Fold increase of CTL

Agegroup

Below 60
(N=
14)

(Mean± SD)

79 ± 22

7913 ±5168

3,818 ±1,894

3,818 ±1,894

7,209

± 3,837

1,025

± 596

Above 60
(Mean± SD)

(N=15)

91 ± 12

9629

± 5958

3,271

± 2,240

56 ± 17

6,093 ±3,846

1,403 ±1,279

P value

0.086

0.414

0.482

0.037*

0.441

0.315

Stage

III

(Mean± SD)

(N=14)

81 ± 23

8,412 ±6,330

3,426 ±1,782

66 ± 15

8,259 ±3,370

1,091 ±522

IV
(Mean± SD)

(N=15)

90 ± 12

9,163 ±4,932

3,637

± 2,352

59 ± 20

5,113 ±3,666

1,342 ±1,325

P value

0.224

0.726

0.786

0.328

0.023*

0.505

Cancer metastatic

Yes

(Mean± SD)(N=13)

89 ± 13

8,280 ±4,687

3,181 ±1,848

58 ± 21

4,858 ±3,893

1,248 ±1,379

No

(Mean± SD)(N=16)

82 ± 22

9,223 ±6,301

3,823 ±2,238

66 ± 14

8,073 ±3,178

1,198 ±618

P value

0.297

0.648

0.405

0.265

0.025*

0.905

# Values were calculated from a total of 29 infusions

Point 9: The Discussion paragraph mostly repeats results and this is not necessary. 

We did rewrite the discussion paragraph by removing the repeat results (lines 251-253)

The manuscript in spite of few remarks is valuable and interesting. It require a review to be considered for publication.

Reviewer 2 Report

Authors must add references to the manuscript that have been published more recent, preferably within 2-3 years.

The number of patients and lack of a comparative/control group are, in my opinion, the major issues in the present paper. Therefore, the results presented here are not conclusive.

Could PD-L1 levels (or other biomarkers) among patients be a predictive factor for better responses? A discussion about immune-check point mechanisms should be included.

Author Response

Response to Reviewer 2 Comments

Point 1: Authors must add references to the manuscript that have been published more recent, preferably within 2-3 years.

We thank the reviewer's comment. We added 20 references to the manuscript that have been published more recent, including number 3,4,5,6,7,10,13,14,15,19,20,21,22,23,24,28,29,48,49,50.

Point 2: The number of patients and lack of a comparative/control group are, in my opinion, the major issues in the present paper. Therefore, the results presented here are not conclusive.

Could PD-L1 levels (or other biomarkers) among patients be a predictive factor for better responses? A discussion about immune-check point mechanisms should be included.

We agree with the reviewer’s comment. We plan to perform a follow-up study with a control group so that we will have a more accurate conclusion. We have put this comment in the limitation of our manuscript (lines 295-297)

Besides, we added the discussion about the immune checkpoint inhibitors, and the determination of biomarker levels, such as PD-L1, as predictive factors for the better response in the discussion part (lines 297-303).

Reviewer 3 Report

The manuscript “Evaluating the safety and quality of life of colorectal cancer patients treated by autologous immune enhancement therapy (AIET) in Vinmec International Hospitals” by Phuong et al evaluated the safety and effectiveness of autologous immune enhancement therapy (AIET) for colorectal cancer (CRC). It is significant that AIET caused a 14.3 ± 14.1-month increase in mean survival time for CRC patients in Vietnam. This report is very easy to read, well written, interesting and useful for further clinical applications. Accordingly, this reviewer recommends publication.

Author Response

Response to Reviewer 3 Comments

The manuscript “Evaluating the safety and quality of life of colorectal cancer patients treated by autologous immune enhancement therapy (AIET) in Vinmec International Hospitals” by Phuong et al evaluated the safety and effectiveness of autologous immune enhancement therapy (AIET) for colorectal cancer (CRC). It is significant that AIET caused a 14.3 ± 14.1-month increase in mean survival time for CRC patients in Vietnam. This report is very easy to read, well written, interesting and useful for further clinical applications. Accordingly, this reviewer recommends publication.

Thank you for your positive comments on our manuscript.

Reviewer 4 Report

This is a robust study, which brings promising results on the possibility of introducing adjuvant treatment to the classic treatment of CRC, from the perspective of patients with improved survival and quality of life.

The work has scientific soundness, good writing, adequate methodology and supported by relevant statistics. Thus, consider the article fit for publication, if the article as ethical guidelines is mandatory for publication.

Author Response

Response to Reviewer 4 Comments

This is a robust study, which brings promising results on the possibility of introducing adjuvant treatment to the classic treatment of CRC, from the perspective of patients with improved survival and quality of life.

The work has scientific soundness, good writing, adequate methodology and supported by relevant statistics. Thus, consider the article fit for publication, if the article as ethical guidelines is mandatory for publication.

Thank you for your positive comments on our manuscript.